# Mobile vs. Non-Mobile Live-Streaming: A Comparative Analysis of Users Engagement and Interruption Using Big Data from a Large CDN Perspective

**DOI:** 10.3390/s21165616

**Published:** 2021-08-20

**Authors:** Daniel V. C. da Silva, Antonio A. de A. Rocha, Pedro B. Velloso

**Affiliations:** 1Coordenação de Informática, Instituto Federal Fluminense, Quissamã 28735-000, Brazil; dvasconcelos@iff.edu.br; 2Departmento de Computação, Instituto de Computação, Universidade Federal Fluminense, Niterói 24210-330, Brazil; 3LIP6, Sorbonne University-CNRS, 75252 Paris, France; pedro.velloso@lip6.fr

**Keywords:** live streaming, mobile networks, user engagement

## Abstract

Video streaming on the Internet is constantly changing and growing. New devices and new video delivery mechanisms generate huge gaps in the understanding of how video application works. From exploratory research of one among the largest streaming services in Brazil, this work presents a comparison between mobile and non-mobile users, in large-scale lives. This work focuses on metrics such as engagement, interruption, churn, and payload. This work also presents a report-overview of mobile-users, considering the operating system, geolocation, network access, interruption, and engagement. These results might offer potential information for streaming improvement, in addition to serving as a historical mark.

## 1. Introduction

The volume of video reproduced over the Internet has never been higher. The advancement of the overall network infrastructure, deployment of content propagation strategies on decentralized servers, robust improvement of host devices, and implementation of new algorithms (e.g., adaptive bitrate) are some of the main reasons that justify the increase in popularity of this service. All of these technologies, developed for video playback over the network, have also allowed live-streamed events to gain more and more audience, turning video streaming applications as the top first in terms of traffic usage over the Internet, as reported by Sandvine [1] and Cisco [2].

Nowadays, with the current context of social isolation imposed by the COVID-19 pandemic, those numbers became even more impressive, reaching new records not only on the number of viewers, but also in the number of live streams, as listed by Billboard in [3]. Instagram, for example, reported that the occurrence of mobile live streaming has increased by 70% [4]. In time of quarantine, an increase in global network traffic has also been noticed [5,6], in which video streaming has a significant contribution. According to [2], video streaming applications lead the bandwidth consumption and the popularity rank among all Internet systems, without any downward trend for the coming years.

Even with all these impressive numbers and technological improvements, when it comes to live video streaming on the Internet, many gaps are still open, such as: Is there a difference in experience or how users of different types of devices consume video? When it comes to streaming, is there an impact of network access? Or even how to measure the engagement, considering nuances and technology changes and cultural changes in the way video streaming is consumed by users. Furthermore, whereas data from video services belong to large companies, what is the current scenario of this multimedia application?

Some previous works have proposed and presented video analysis on a large scale, such as [7,8,9,10]. These works are very relevant to the research field, but they do not exhaust the topic because they use very regionalized data, or only from the user’s perspective (without the server-side panorama) or even without the scale of users or volume that we have today. Another aspect is that many of these studies focus their efforts on a specific metric. These metrics limitations, for example, are possibly due to the data format that the researchers had access to.

In a previous study [11], we presented a first look at live streaming application, considering a clipping of mobile devices. Such analyses were motivated by the fact that in many countries, smartphones are already the most used device, and not only for Internet access, but also for watching video streaming [12]. In this previous work, we show that the operating system, the network access technology and even the end user geolocation could bring relevant influences on engagement time and quality experimented by the users in live video streaming sessions.

However, some unanswered questions are still open: are they significantly different from the overall behavior and quality of experience for live streaming users when connecting through mobile and non-mobile devices? That is, do these characteristics (operating system, network access, geolocation, etc.) apply only to mobile devices? If so, what are these differences? In addition, to answer those questions, this work has the challenge of establishing an approach to analyze big data from a large CDN, focused on the live video streaming service. Up to our knowledge, the presented analysis is unprecedented in the literature. It shows evidence of important aspects that makes mobile users less failure tolerant than non-mobile users when these two groups have similar experiences.

In addition to using data that almost all streaming service providers do not share, this work is especially challenging due to the difficulty in computing big data and interpreting data within relevant metrics for the community. One of the most important metrics is user engagement in a video streaming session, an almost unanimous measure in studies of this nature, which is vital for services, content creators, advertisers, and users.

This work also brings a section of general information about the streaming scenario, considering grouping users in several different criteria, such as geographic distribution, network access, operating system, engagement bands, and reproduction problems. This part of the text, which can be seen as a report, presents general data of the studied streaming system and has great potential for use in other studies, future implementations, and even infrastructure investment decisions.

This paper is organized as follows: In the next section, we highlight the state-of-the-art for related works. In Section 3, we detail the methodology used in this work, which includes the description of the dataset and the considered metrics of analyses. In Section 4, we present a comparative analysis of the fundamental metrics in mobile non-mobile devices. Then, in Section 5, we consider the operation of mobile devices in different perspectives and investigate the characteristics of those users. Furthermore, finally, we present the final remarks in Section 6.

## 2. Related Work

In the literature we find three main categories of related work. In the first category, mobile-user perspective, two works stand out [9,13]. Dimopoulus et al. [13] have developed a framework to detect the source of failures and the reduction of QoE (Quality of Experience) of user video streaming on mobile devices. The framework consists of three measurement points: two on the end-points and one on the wireless router. Furthermore, it considers interruption (stall) and throughput (bit-rate) metrics. In [9], the authors evaluated Youtube video transmission looking at the cellular network. Their purpose was to quantify the impact in QoE considering parameters, such as browser, transport protocol, bandwidth and signal. Both works, [9,13], use video on demand (VOD) data collected from the client-side, a different approach from our work.

About large-scale transmission, we have selected four works. In [7], the authors present the first look at video transmission in large-scale for the cellular network and bring a pertinent analysis of transport and network link for vídeo streaming. They bring important results, such as: users have low viewing duration/engagement (playbacktime) on long videos. These are important results, but it still does not differentiate between mobile and non-mobile users. In paper [14], the authors aim to study the data flow of the team’s supporters present in the stadium during the Super Bowl, which is among the world’s most popular sporting events. This work shows that a noteworthy part of the traffic came from video streaming applications, possibly from transmissions of the match itself. It was considered access from mobile devices using Wi-Fi and cellular networks. Ahmed et al. [15,16] studied the QoE on large-scale live video streaming transmissions. In [15], they analyzed the mean rate, the buffering rate, and its variation, using CDN’s log records from an HTTP based server. In [16], the authors compared the users’ engagement (viewing duration) with their QoE in the live-streaming broadcast of the Oscar 2015. The work considers aspects such as buffering and adaptive bitrate.

To the last group, mobile-live, three works were selected [17,18,19]. Salamatian et al. [17] analyzed live-video transmission from a hybrid P2P/client-server service to mobile and non-mobile. This work considered cellular network and Wi-Fi, as well as hardware differences. The authors also presented metrics such as viewing duration (engagement), abandonment rate, user activity and geographical distribution. They concluded that session failure rates on 3G are higher than on Wi-Fi, for different reasons. Although failures in the 3G session are due to the poor access network, failures in the Wi-Fi session are usually a consequence of loss of interest or connection problems. Based on the Periscope platform, Siekkinen et al. [18] proposed to evaluate the quality of the live video transmission session from the perspective of mobile devices. Developed by the authors, a monitoring system performed the experimental sessions to capture data for the proposed metrics. In those experimental transmissions, they collected data on video quality, battery usage, latency, etc. They present several thresholds between quality and problems in the sessions, and also a correlation between the watching time and stall rate. Furthermore, finally, in [19], the authors propose a way to evaluate live video sessions of 4G (LTE—Long Term Evolution) cellular network users. This work presents and proposes an implementation of ITU-T P.1203, which defines recommendations for monitoring algorithms of the life cycle of a media session, from the network point of view. The authors also present a comparison, in a real and controlled environment, of the evaluation technique with progressive download and with an adaptive strategy.

All of these works presented above are related to the transmission of video over the Internet. However, our approach differs from these and other studies in the area, mainly because it considers and classifies mobile and non-mobile users, large-scale scenarios, and use data generated by the streaming service server.

## 3. Background and Methodology

The main purpose of this work is to make an exploratory evaluation of live video, comparing mobile and non-mobile devices, using data from a large CDN. To achieve this purpose, this paper uses data from Globo.com [20], which is the biggest TV channel and video streaming company in Brazil.

### 3.1. Architecture and Dataset

Globo.com uses NGINX [21], an open-source web service, to manage its HTTP servers. NGINX controls the streaming video service via the Live Streaming HTTP (HLS) protocol, which is an HTTP-based adaptive bitrate streaming protocol. This infrastructure, depicted in Figure 1, manages the generation, the segmentation and the transmission of video content to connected users. All contents are encoded with different qualities, and then segmented for broadcast. HLS defines video segmentation into small chunks.

The dataset used in this paper comes from the broadcast server log that records information of each video chunk sent to users. At the end of each day, the server wrote these log messages into several plain/text files. Each log file contains millions of records and has up to 2 GB and ranging from 50 MB to 350 MB in compressed format. Furthermore, given the specifics of the data, the authors chose to use scripts written in Python for information processing and use a relational database for persistence.

The dataset contains the following fields: (i) user ID, (ii) IP address, (iii) user agent, (iv) date-time for each chunk, (v) payload of each chunk (bytes), (vi) operating system, (vii) network access (whether it is mobile or non-mobile), (viii) geolocation, (ix) transmission channel, etc. So, from this information, it was possible to retrieve/recreate the users’ session data from these log files. For example, it was possible to calculate the time of entry and exit of users, total payload, interruption, etc.

### 3.2. Metrics

From the dataset of Globo.com’s video server, it is possible to extract relevant information from the recorded streaming sessions. In this process, data is interpreted based on some metrics. This step is important to (i) establish the potential for data interpretation and (ii) ensure that it is compared with other studies that were structured based on different data.

**Viewing Duration Time**: this metric shows the duration of a user session, that is, the time a user keeps watching a given live transmission. This measure depicts a user engagement in a given transmission. Viewing duration is, by far, the most used metric in similar studies, so considering it is essential for comparisons between studies. This approach of using view duration as engagement is widely used in the literature, in studies such as [7,16,17].

**Interruption and Reentry**: an interruption is a failure to play the video on the user’s device. From the streaming server log, it can be characterized by a set of packages missing during a user session. A related concept is that of reentry. A reentry is defined as any further arrival of the same user in the transmission session for the same video, after an interruption. Weighing interruptions as a metric is imperative to understand the quality of experience perceived by users of a live streaming video session. It is essential to point out that the occurrence of interruptions is directly associated with the QoE.

**Throughput**: a user’s throughput is the result of the sum of all bytes received in each video chunk during a session divided for that user’s time in the system. The transmission rates practiced by the Globo.com are the primary comparison of this metric. Throughput is a relevant metric because it is able to clearly point out whether or not a given user has practiced a transfer rate compatible with the video chunk size.

**Arrivals and Departures**: this metric defines the number of users logged in at a given time interval for a specific live video streaming session. It is possible to calculate the total number of simultaneous users throughout the transmission, using the timestamp of each chunk recorded in the log. Therefore, this measure is very useful to understand crowd scenarios, and to consider aspects of engagement related to the content.

### 3.3. Interruption

In a first look at the Globo.com data, it was possible to notice some peculiarities, for instance, a significant number of chunks with no payload, or with a considerable interval between chunks. These peculiarities could indicate that users might have experienced interruptions. It means that the player was not able to reproduce the live-video, at some point during the session. These interruptions have a significant impact on users’ engagement in live transmissions since it’s the crucial premise that relies on following an event in real-time. A good example is a sports event, in which any interruption might reduce user engagement and might imply missing a critical moment of the game.

Thus, characterizing user’s interruptions is crucial to understand it’s behavior. Nonetheless, identifying interruptions is not a trivial task. Interruptions can occur due to network problems, software issues at the client, or simply because the user is no longer interested in continuing watching the transmission. Therefore, we investigate the best approach to identify interruptions, considering the information available in our dataset, and we came up with three different criteria (including some combinations) to define interruptions.

**No interruption**: the viewing duration of a user starts when a user receives the first video chunk, and it ends with the last one registered, and thus, it does not recognize reentry. This reading could be used as a baseline of comparison with the other criteria.**Payload issue**: this criterion implements clause (a): if there are at least three consecutive video chunks with no payload (zero bytes) from a user, we consider it as an interruption of the video. Therefore, any future video chunks of the same user considered as reentry in the broadcast session of that video and count as a new user session.**Missing chunks**: Clause (b): Since the NGINX video segmentation service suggests 3 to 10-s video blocks, clause (b) considers an interruption when a specific user is not receiving information for a given time interval. If, after this specified interval, the user gets video chunks again, it should be considered a system reentry (and count as a new user session). Several time intervals have been experimented to accomplish this characterization.**Combination A + B**: this criterion implements the clauses (a) and (b), previously described.**Combination A + B + zero payload**: this implements clauses (a) + (b) and also ignores users that have not received a single byte (all chunks with zero payload).

## 4. Comparative Analysis

Our first goal in this paper is to compare the characteristics of the fundamental metrics previously presented in two distinct scenarios: mobile devices and non-mobile devices. To achieve this objective, we selected two popular events of 2016: (i) the impeachment admissibility session of former president Dilma Rousseff and (ii) the opening of the Rio2016 Olympic Games.

### 4.1. Interruption and Reentry

The first step before performing our comparative analysis was to evaluate the previously described criteria for identifying interruption. We have applied all combinations defined in Section 3. Hence, through the number of reentries, it is possible to evaluate the volume of interruptions, and the results show that clause(b) was the only one relevant in this aspect.

We evaluated several distinct time intervals for clause (b), as show in Table 1. These time slots represent the amount of time a given user spent without receiving new video snippets. Table 1 also shows the total value of unique users and the number of user sessions considering reentries with different interruption intervals.

We can see from Table 1 that regardless of the value of the interruption time interval, in the Impeachment session, the non-mobile had a significantly higher number of interruptions. This difference in the number of interruptions is not proportional to the duration of the session, namely, the duration of the Impeachment session is 10 h, while that of the opening ceremony of the Olympics is 5 h.

Furthermore, as we increase the value of the interruption time interval, the proportion of the number of interruptions increases, especially in the Impeachment broadcast section. It means that the Impeachment broadcasting has more prolonged interruptions than the Opening of Rio 2016 Games, considering both mobile and non-mobile devices. To explain this data, first, we need to take into account the duration of the evaluated sessions. Secondly, it is necessary to consider the content: the Impeachment session was a very long voting process, and that interruption time did not hinder the understanding of the result. On the other hand, the opening of the Olympic Games featured several performances, attractions, and concerts.

Figure 2, Figure 3, Figure 4 and Figure 5 show the classification of users according to the number of interruptions, from highest to lowest. This result corroborates the previous reading, showing that both mobile and non-mobile Impeachment viewers have a more extended number of interruptions. In the figures, the users with the worst results, that is, those with the most interruptions, are in the first area, which have a significant number of users with more than ten interruptions, and includes users with alarming numbers of up to hundreds. Considering these users in both broadcast sessions, it is evident that non-mobile users have a much larger number of interruptions. This first difference between users points out that non-mobile users are more tolerant since they remain in sessions despite the obvious problem. This fact may perhaps be justified when considering that non-mobile users may have no concerns about battery use or data traffic, as opposed to mobile users on what costs of communication and mobile coverage are major issues.

Table 2 was calculated to broaden the understanding of this disparity between mobile and non-mobile users. This table shows the number of (i) unique users (not considering interruption), (ii) the number of users’ sessions (considering reentries as new users—through the clause-b), and (iii) presents a perceived increase in users comparing these two criteria. From the data presented in the table, it is possible to notice that the operating systems of mobile devices have no general correlation with the occurrence of interruptions.

### 4.2. Viewing Duration and Engagement

The most relevant metric is possibly the Viewing Duration because it represents the actual playback time of a user, and thus directly represents their involvement in the broadcast session. Figure 6, Figure 7, Figure 8 and Figure 9 illustrate the complementary cumulative distribution function of users’ total playback time, that is, the viewing duration, segmented by the number of interruptions suffered by the user. These interruptions were perceived using Clause B with a 20 s interval, already covered and explained in the previous section.

In both broadcast sessions, the most obvious difference between mobile and non-mobile users appears in the range of those experiencing ten interruptions or more. Another noticeable difference is that mobile user engagement is lower than non-mobile devices, even among those with high interruptions. Would mobile users be less engaged and less tolerant to failure?

In Dilma’s first Impeachment broadcasting session, illustrated in Figure 6 and Figure 7, excluding the users with more than ten interrupts, the other curves behave very similarly when we compare the engagement, regardless of device type. At the same time, the difference in the number of interruptions experienced by users of the Impeachment session, when compared to the users of the Olympics may be explained by users who would not be willing to stay connected for hours but wanted to be updated on the voting results.

At the opening of the Rio2016 Olympic Games, illustrated in Figure 8 and Figure 9, it is possible to observe that mobile users with up to 1 interruption usually had a longer viewing time, while users with two or more interruptions had below-average viewing time. Interestingly, in the same broadcast, non-mobile users without interruption had the least significant engagement. These facts reinforce the idea that non-mobile users are more resilient and fault-tolerant, while mobile users are more concerned with using device resources (such as data plan and battery). It is worth mentioning that other issues may influence mobile playtime metrics. They are expected to have more limitations on their ability to play videos for longer periods, justified, for example, by battery life and limitation of mobile data transmission.

### 4.3. Arrivals and Departures—Users in the System

Each of Figure 10, Figure 11, Figure 12 and Figure 13 presents three curves: the black line with unique users who first logged-in and remained without interruption/reentry. The second curve indicates the users on a reentry, that is, those who have already been interrupted and returned to the transmission. Finally, the green line is the total users added up.

The general comparison that we can make between the two user classes (mobile and non-mobile) is on their volume of simultaneous users. Non-mobile users users are still much more numerous, while a similarity between these two classes is that the curve’s shape shows that the overall engagement behavior is similar between each event. This similarity is noticeable even in a transmission session problem, for example: (i) between 3 h and 5 h in the Impeachment Figure 10 and Figure 11 and (ii) just after 4 h in the Olympic Games Figure 12 and Figure 13. Considering that might be some transmission system problem, it is noticeable that mobile users have a slower recovery than non-mobile.

Looking at the numbers of the users connected to the broadcast of the Opening of Rio2016 Olympic Games, we can observe that most users have started watching the broadcast during the last 90 min, just at the end of the Olympic delegation parade and the returning of shows. A last interesting observation about the Impeachment session is that between the 3rd and the 5th hours, the number of users on a reentry surpasses the users with no interruption, reinforcing the previous reading about users not willing to follow all hours but interested in timely following votes, noticeable in both evaluated user classes.

### 4.4. Throughput

This metric measures the user throughput, calculated by summing all the bytes received by the time a given user remained engaged. In Figure 14, Figure 15, Figure 16 and Figure 17, each dot represents a user. The expected transfer rate of the servers from Globo.com, as suggested in [8], are represented by straight lines, as well as the linear regression function for the throughput for all users. Before beginning the direct interpretation of these abstractions, it is essential to note that no user is above the supposed higher transmission rate. This fact is explained by eventual retransmissions (taking into consideration that HLS uses TCP as a transport protocol, which justifies these possible retransmissions), or even users who made use of the “rewind” function, that is, they returned the transmission gradually to review some part of the video.

On the other hand, many users got below the supposed lower baud rate. Most mobile users below 100 kBps stayed for a short time. In contrast, a considerable part of non-mobile users remained in the system for a long time, despite the apparent problem, another fact that asserts a low-tolerance for problematic sessions that mobile users have.

## 5. Live-Streaming Scenario

This section evaluates the scenario of popular live streams for mobile devices, in regard to characteristics such as the operating system, the network access, and geolocalization. In this section, we analyze the Dilma Impeachment broadcasting sessions (i) House of representatives, (ii) Senate, and (iii) Judgment Impeachment sessions, the (iv) 2016 Eurocup semi-final match between Portugal and Wales, and (v) the Opening Ceremony of the Rio 2016 Olympic Games, which are among the most popular sessions of 2016, summing up to over 620,000 unique users of mobile devices.

### 5.1. General Settings

Table 3 shows the distribution of the operating system and network access for users in the five regions of Brazil and abroad. We identified two operating systems: Android and iOS (regardless of their versions) and classified the Internet access as (i) mobile network, such as 3G, 4G, or equivalent, which is referred to as 3G in Table 3, and (ii) wired network. Since the vast majority of mobile devices have IEEE 802.11 network adapters, it was labeled Wifi.

In Table 3, the values between parenthesis represent the distribution of the users of all transmissions. The Southeast region concentrates the majority of users, in counterpoint to the North of Brazil. It is worth emphasizing that the share of users outside Brazil is relevant, and is only smaller than in the Southeast. Furthermore, in Table 3, it is possible to notice that the South, Northeast, North, and Central-West regions are similar in the presented characteristics, approximately 1/3 of users use iOS and 2/3 Android, and more than 80% were using a wired connection (most likely through an Wifi access point). In all regions, Android is the operating system most used by live streaming users. The opposite was verified for users outside the Brazilian territory. Only the Southeast and users from abroad have surpassed the 30% barrier of users using a mobile network, which is a significant number, but it shows that a large proportion of users, although using mobile devices, these users are not using mobility.

### 5.2. Playback Time

Playback time is an important metric when it comes to video streaming, as it is directly related to user engagement. Figure 18 shows the distribution of users’ playback time of the five analyzed sessions in this study. The *y*-axis represents the duration of the visualization (in seconds), represented in a logarithmic scale. Each of the sessions, identified in the *x*-axis and by different colors, presents a box diagram with values of extremes and quartiles. The black square represents the average of the viewing duration of the users of each session.

It is possible to notice that most users have a short viewing duration, because, in almost all sessions, half of the users will hardly exceed 1 min, the exception is the Eurocup, which reaches 8 min. Besides, the average of users’ playbacktime in all events are between 4 min and 19 min, which does not represent a high value compared to the duration of the broadcast sessions, all on a scale of hours.

As described by media [22,23,24,25], large video platforms have, in the recent past, changed their engagement perception policies. These changes included considering users within seconds for commercial purposes. This change inflated user perceptions between 80% and 900%, according to journalistic reports. Thanks to the publication of these data, it is possible to confirm the user engagement values presented by this study, although it seems counterintuitive this volume of users with such low viewing duration.

### 5.3. Number of Interruptions

The number of interruptions is a metric that affects the quality of the user experience. In our definition, previously presented, when a user does not receive video chunks for more than 20 s, an interruption is characterized.

Figure 19 shows the distribution of the five sessions analyzed in this study. Similar to the previous graph, changing only the playback time for the number of interruptions. It is evident in this illustration that more than 25% of users of all sessions do not have any interruption, and half have suffered only one. The first difference is in the third quartile, with numbers ranging from 1 to 3. Considering that all events had a long duration, these few interruptions may be tolerable by users.

The real problem with the interruptions is presented in the maximum values of the distribution, as can also be seen from Figure 19. For example, the highest value, that is, 523 interruptions of the session of admissibility of the Impeachment in the Brazilian Senate, and considering that each interruption had at least 20 s, indicates that this user had more than 2 h of interruption, which is a high value, even taking into account that the sessions under analysis are very long.

## 6. Conclusions and Future Works

This paper aimed to compare mobile and non-mobile users and further analyze mobile users from the defined parameters and metrics. Mobile and non-mobile users have many similarities, but the first group is far less tolerant of potential problems. The data also suggests that mobile users are more concerned with device features and endures limitations in the data plan and mobility. While non-mobile users are much more likely to stay in problematic or even uninteresting broadcast sessions. It is imperative to point out that the reading of these results must take into consideration the nature of the sessions studied.

In addition, we can summarize the following results discussed in this paper: (i) Among users with alarming numbers of video playback interruptions, non-mobile users have much higher maximum numbers; (ii) Proportionally, mobile suffer more interruptions; (iii) Operating system has no apparent impact on interruptions experienced by mobile users; (iv) Considering users with many interruptions: mobile users have a low viewing duration, when compared to non-mobile users and with mobile users with fewer interruptions; (v) Non-mobile users have more play time; (vi) The number of interruptions suffered by users has more relevant consequences for mobile ones. Furthermore, this indicates that they are less tolerant of this type of failure. Furthermore, in contrast, non-mobile users are more resilient and fault-tolerant; (vii) The general engagement pattern of the session audience is very similar between mobile and non-mobile; (viii) Transmission noise generates more degradation in mobile users, which take much longer to recover from this type of event; (ix) Mobile users with low throughput remain for much less time than mobile users within the minimum transmission rate and non-mobile users, in general; (x) The content has enormous potential for influencing the engagement pattern, and any study on the matter should consider this.

In the end, the most in-depth assessment of mobile users shows that mobile users are still a minority, and among those who use mobile Internet access (like 3G, 4G) are an even smaller share, despite widespread access to mobile phones. These data were compared to the information published from other video platforms, and despite the intriguing data presented, these are corresponding.

Despite the analyses presented in this paper consisting of five popular event session transmissions, we have also analyzed other sessions of lower/same popularities. It is worth mentioning that the option of limiting to those is because they are the most popular sessions so far, but others of lower popularities had mostly similar conclusions.

For future works, we consider that two fronts are really important: (i) to understand the comparative analysis of users engagement and interruption using big data from a large CDN perspective for video on demand services. For instance, most of live-streaming providers (e.g., Globo.com) also offer a VoD service. How different/similar are the analyses for such kinds of users? Furthermore, (ii) to define new approaches that use features extracted from the dataset and use machine learning algorithms with the aim to predict user’s engagement and interruption of a video service (live-streaming or video over demand). We understand that such a solution has a great potential to help video content providers on dimensioning network infrastructure and to properly adapt users service. 

## Figures and Tables

**Figure 1 sensors-21-05616-f001:**
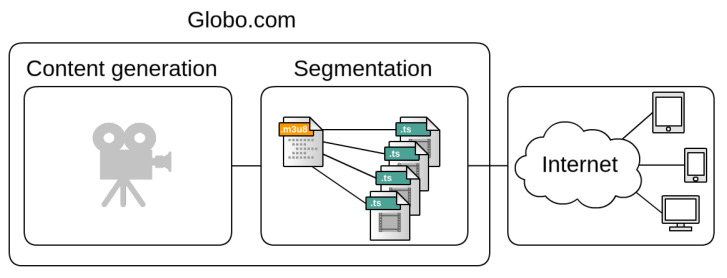
Globo.com infrastructure.

**Figure 2 sensors-21-05616-f002:**
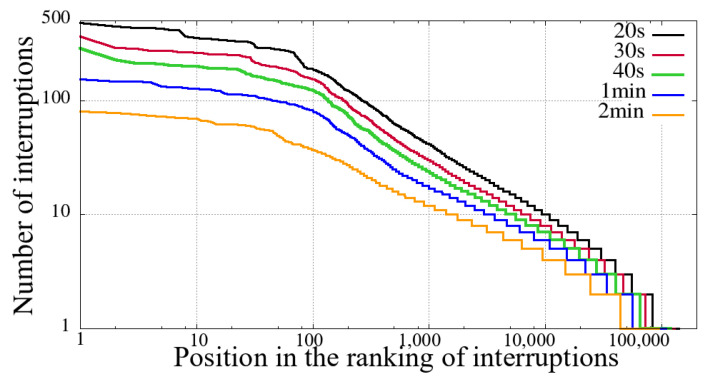
Interruption ranking from users of the Impeachment start session—*Camara dos Deputados* for different sizes of interruption—mobile devices.

**Figure 3 sensors-21-05616-f003:**
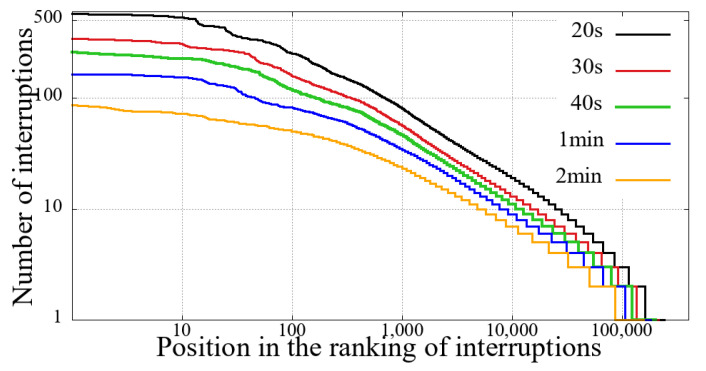
Interruption ranking from users of the Impeachment start session—*Camara dos Deputados* for different sizes of interruption—non-mobile devices.

**Figure 4 sensors-21-05616-f004:**
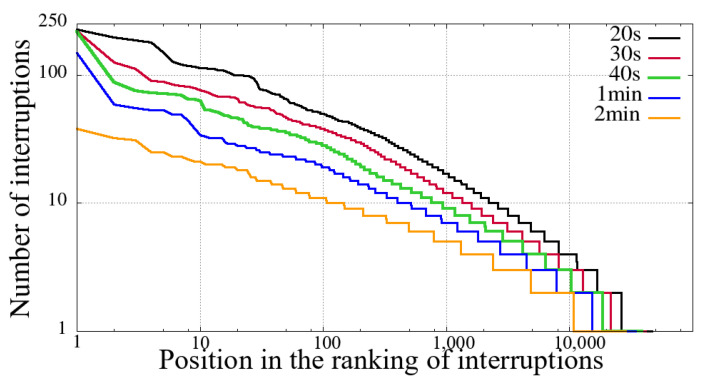
Interruption ranking from users of the Opening Rio 2016 Olympic Games for different sizes of interruption—mobile devices.

**Figure 5 sensors-21-05616-f005:**
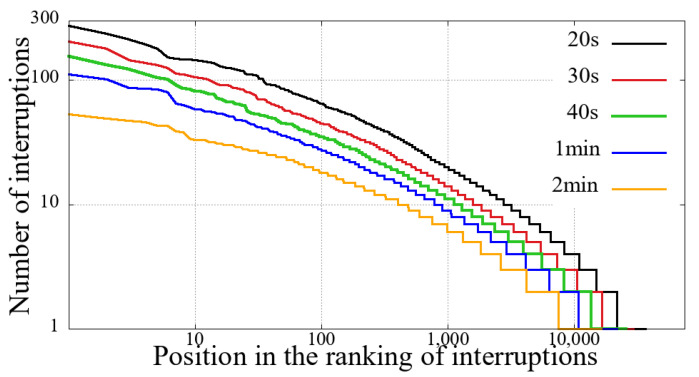
Interruption ranking from users of the Opening Rio 2016 Olympic Games for different sizes of interruption—non-mobile devices.

**Figure 6 sensors-21-05616-f006:**
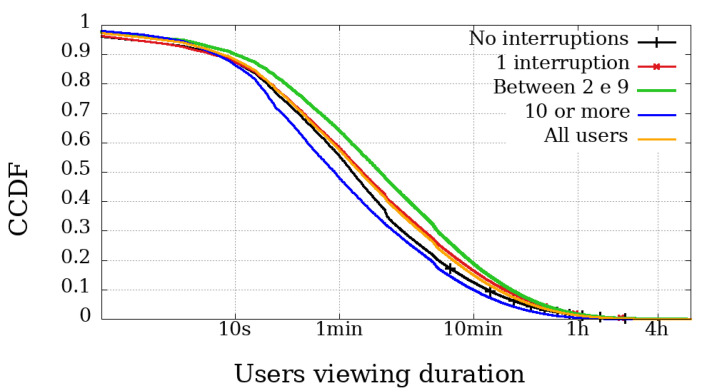
Complementary cumulative distribution function of the viewing duration from users, grouped by the amount of interruptions experienced in the transmission session of the Impeachment start session—*Camara dos Deputados*—mobile devices.

**Figure 7 sensors-21-05616-f007:**
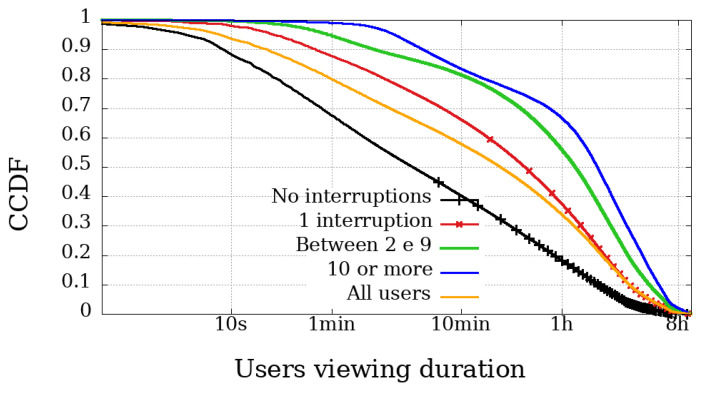
Complementary cumulative distribution function of the viewing duration from users, grouped by the amount of interruptions experienced in the transmission session of the Impeachment start session—*Camara dos Deputados*—non-mobile devices.

**Figure 8 sensors-21-05616-f008:**
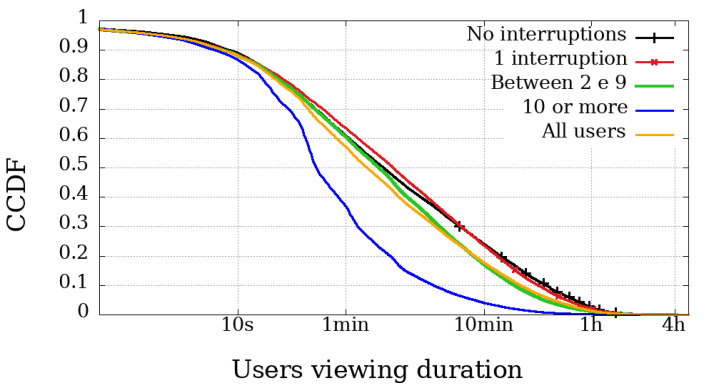
Complementary cumulative distribution function of the viewing duration from users, grouped by the amount of interruptions experienced in the transmission session of the Opening Rio 2016 Olympic Games—mobile devices.

**Figure 9 sensors-21-05616-f009:**
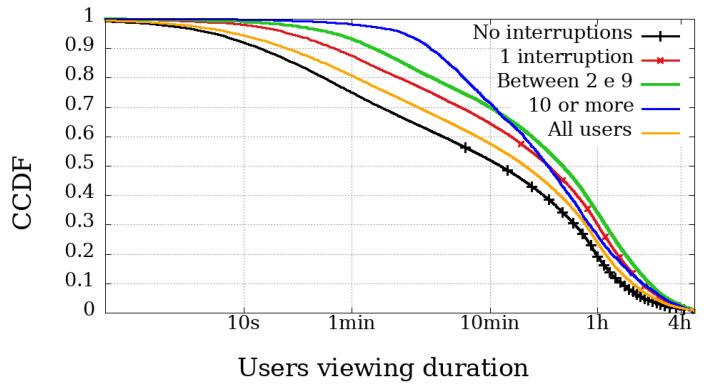
Complementary cumulative distribution function of the viewing duration from users, grouped by the amount of interruptions experienced in the transmission session of the Opening Rio 2016 Olympic Games—non-mobile devices.

**Figure 10 sensors-21-05616-f010:**
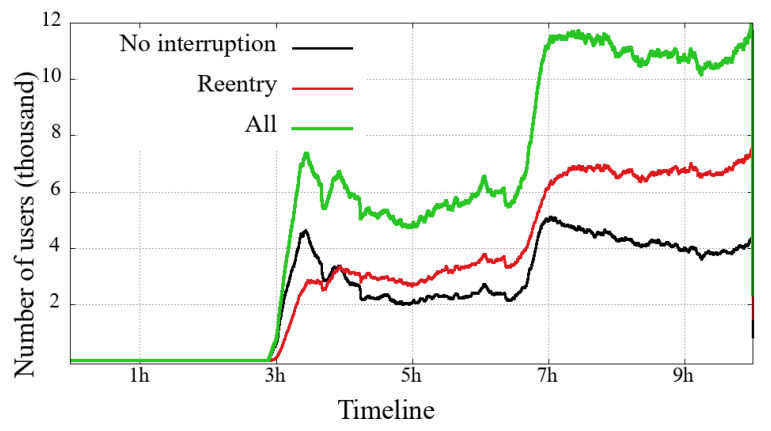
Mobile users on the system in the broadcasting session of Impeachment start session—*Camara dos Deputados*.

**Figure 11 sensors-21-05616-f011:**
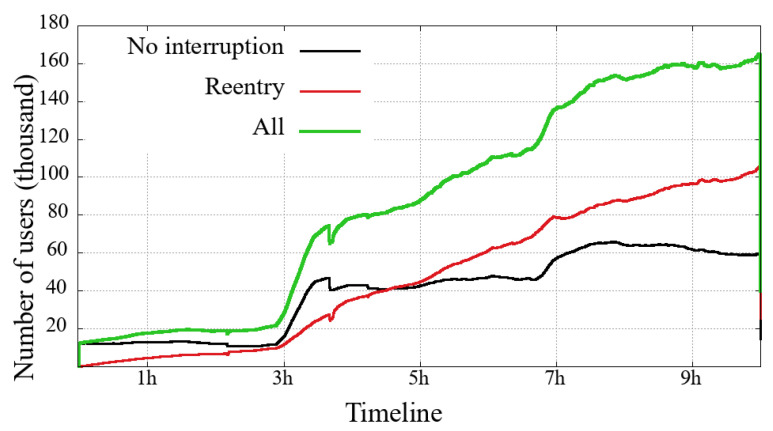
Non-Mobile users on the system in the broadcasting session of Impeachment start session—*Camara dos Deputados*.

**Figure 12 sensors-21-05616-f012:**
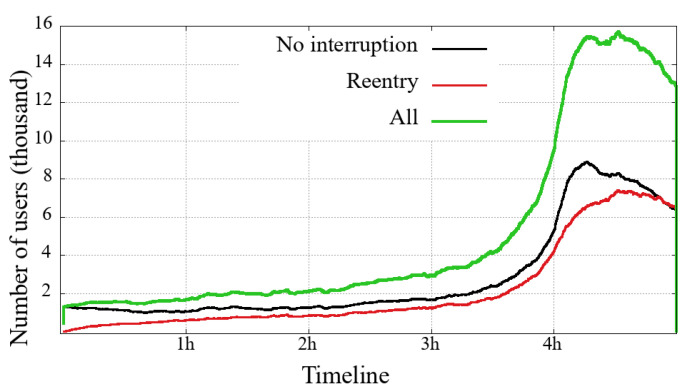
Mobile users on the system in the broadcasting session of Opening Rio 2016 Olympic Games.

**Figure 13 sensors-21-05616-f013:**
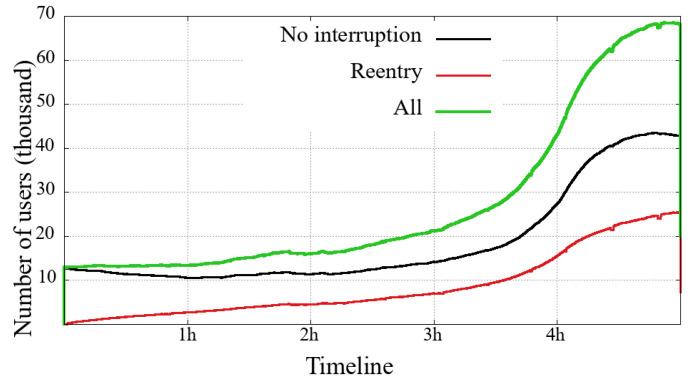
Non-mobile users on the system in the broadcasting session of Opening Rio 2016 Olympic Games.

**Figure 14 sensors-21-05616-f014:**
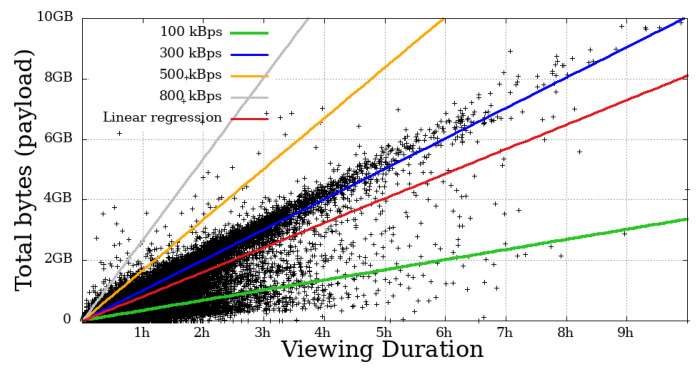
Bytes throughput per second of mobile users on the system in the broadcast Impeachment start session—*Camara dos Deputados*.

**Figure 15 sensors-21-05616-f015:**
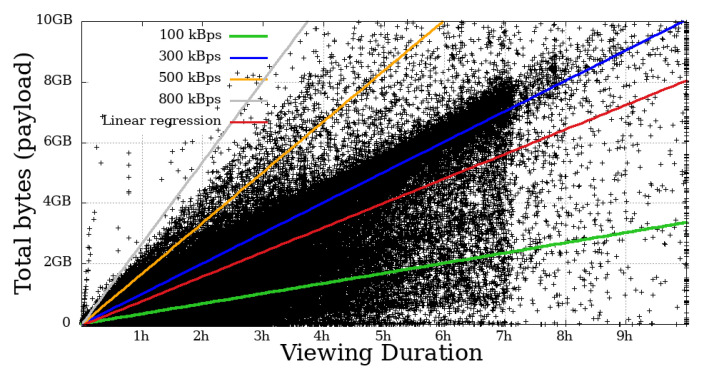
Bytes throughput per second of non-mobile users on the system in the broadcast Impeachment start session—*Camara dos Deputados*.

**Figure 16 sensors-21-05616-f016:**
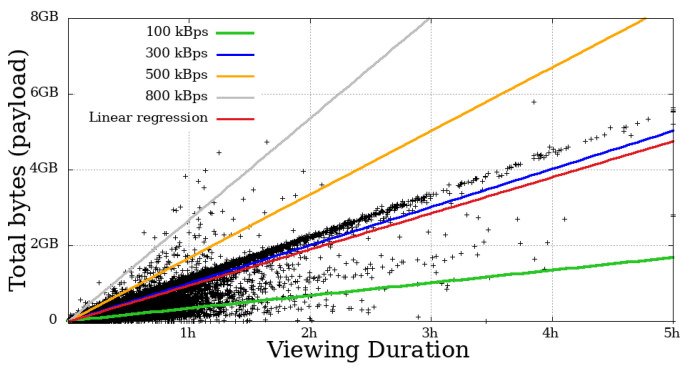
Bytes throughput per second of mobile users on the system in the broadcast Opening Rio 2016 Olympic Games—mobile.

**Figure 17 sensors-21-05616-f017:**
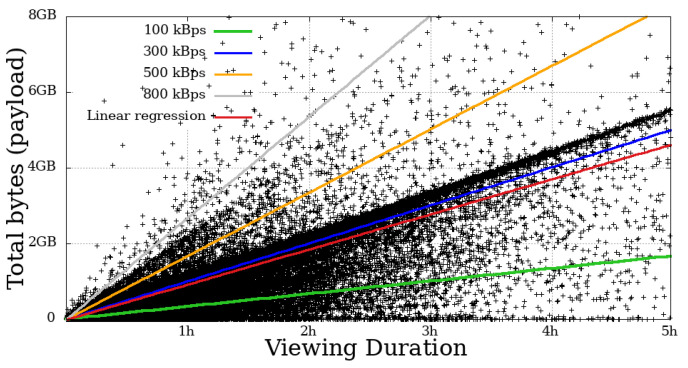
Bytes throughput per second of non-mobile users on the system in the broadcast Opening Rio 2016 Olympic Games.

**Figure 18 sensors-21-05616-f018:**
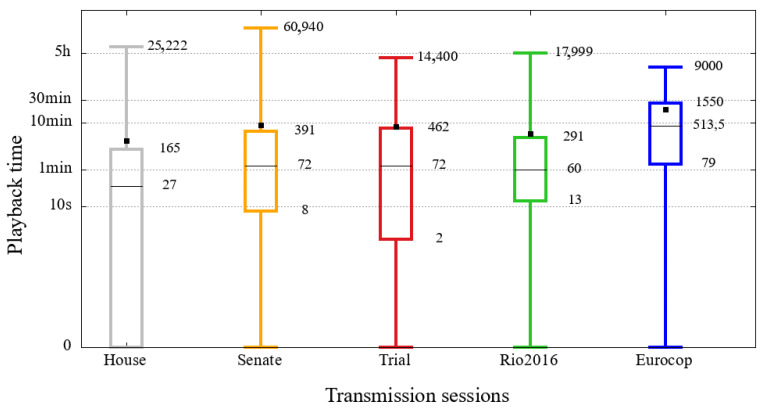
Distribution of user playback time in the opening session of the Rio 2016 Olympics, the 2016 Eurocup semi-final between Portugal and Wales and the House of representatives, Senate and Judgment Impeachment sessions.

**Figure 19 sensors-21-05616-f019:**
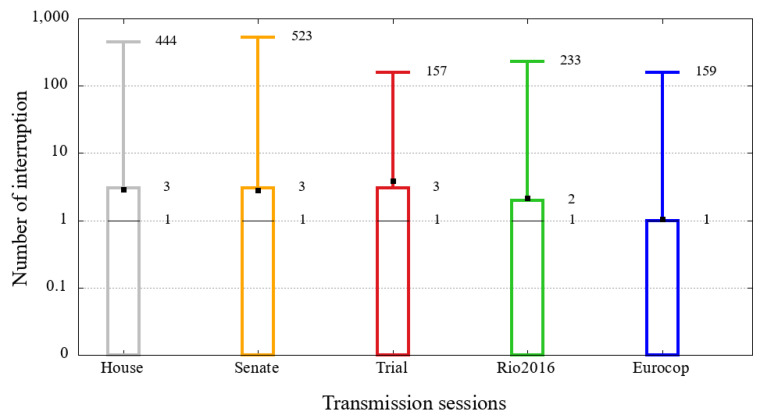
Distribution of the number of interruptions in the broadcasting session of the opening of the Rio 2016 Olympics, the Eurocup 2016 semi-final between Portugal and Wales and the House of representatives, Senate and Judgment Impeachment sessions.

**Table 1 sensors-21-05616-t001:** Total session interruptions, considering Clause B with different time intervals.

Sessions	Olympic Games		Impeachment	
	Mobile	Non-Mobile	Mobile	Non-Mobile
Unique entries	105,964	310,979	542,856	1,054,614
**Total Interruptions—Clause B**				
Interval 20 s	154,906	151,571	576,299	1,254,037
Interval 30 s	116,034	102,032	452,898	891,665
Interval 40 s	93,314	78,692	380,084	744,728
Interval 1 min	72,760	59,965	302,521	596,000
Interval 2 min	50,104	39,128	216,101	436,572

**Table 2 sensors-21-05616-t002:** Increased user perception considering Clause B—20 s.

Sessions		Rio2016		Impeachment	
		Users		Users	
Unique users	Android	58,567	13.69%	333.102	20.85%
	iPhone	45,063	10.53%	153.842	9.63%
	iPad	13,156	3.08%	55.912	3.50%
	Non-Mobile	310,979	72.70%	1054.614	66.02%
		Users		Users	
Clause B-20	Android	140,265	18.22%	1057.874	24.17%
	iPhone	108,422	14.08%	507.696	11.60%
	iPad	26,640	3.46%	196.790	4.50%
	Non-Mobile	494,640	64.24%	2615.255	59.74%
Increased perception	Android	139.49%		217.58%	
	iPhone	140.60%		230.01%	
	iPad	102.49%		251.96%	
	Non-Mobile	59.06%		147.98%	

**Table 3 sensors-21-05616-t003:** Combined percentage of Internet access and operating system by region.

Southeast (56.12%)		3G	Wifi	
Android	17.76%	41.76%	59.52%
iOS	14.92%	25.56%	40.48%
	32.68%	67.32%	
South (10.81%)		3G	Wifi	
Android	8.60%	58.86%	67.46%
iOS	5.66%	26.88%	32.54%
	14.26%	85.74%	
Northeast (11.15%)		3G	Wifi	
Android	6.61%	62.99%	69.59%
iOS	4.75%	25.66%	30.41%
	11.36%	88.64%	
North (2.56%)		3G	Wifi	
Android	9.29%	62.18%	71.46%
iOS	6.19%	22.35%	28.54%
	15.48%	84.52%	
Central-West (7.72%)		3G	Wifi	
Android	9.68%	57.85%	67.53%
iOS	6.75%	25.72%	32.47%
	16.42%	83.58%	
Out of Country (11.64%)		3G	Wifi	
Android	13.01%	23.08%	36.09%
iOS	18.34%	45.57%	63.91%
	31.35%	68.65%	
Total (100.00%)		3G	Wifi	
Android	14.13%	45.57%	59.70%
iOS	12.33%	27.97%	40.30%
	26.46%	73.54%

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
