# Peer review of "Mobile vs. Non-Mobile Live-Streaming: A Comparative Analysis of Users Engagement and Interruption Using Big Data from a Large CDN Perspective"

_sensors, 2021, doi:10.3390/s21165616_

Round 1

Reviewer 1 Report

The paper analyzed a dataset of live video from a large Brazilian CDN, comparing mobile and non-mobile devices. The authors use metrics such as engagement, interruption, churn, and payload. In general, the paper is interesting and well organized.

Some questions could be better discussed in the text, especially the counterintuitive: the user engagement, for example. Why is the viewing duration so low? There are other different events in the dataset that could be analyzed? Is this a characteristic of “big” events such as the impeachment and the Olympics? Maybe smaller or regular content (such as movies) could bring new results and insights. The last paragraph of the conclusion could be more discussed (possible reasons for the intriguing data presented).

Some information about how the authors extract the data, programming language, etc could be given. It could be interesting if the dataset (or part of it) could be shared with the community.

The figures of the paper are of low quality and may be improved. The font size of the figure captions is too big. Figures 18 and 19: what is “House of representatives”, “Senate”? It is not clear. 

There are some minor typos in the text. Ex: Line 311: mobile users have a slower recovery than de non-mobile

Author Response

First of all, we thank you all for all the fruitful comments that certainly helped us to improve our paper. We have carefully considered all the remarks and suggestions made by the reviewers. We report below our answers regarding each remark raised by the reviewers and how we have addressed each remark in the revised version of the paper. It is worth mentioning that all the changes considered in the manuscript are marked in red, as suggested in the response letter, "such that changes can be easily viewed by the editors and reviewers". 

Responses to reviewers

=================================================================

Reviewer #1: The paper analyzed a dataset of live video from a large Brazilian CDN, comparing mobile and non-mobile devices. The authors use metrics such as engagement, interruption, churn, and payload. In general, the paper is interesting and well organized.

[Point 1] (a) Some questions could be better discussed in the text, especially the counterintuitive: the user engagement, for example. Why is the viewing duration so low? (b) There are other different events in the dataset that could be analyzed? (c) Is this a characteristic of “big” events such as the impeachment and the Olympics? (d) Maybe smaller or regular content (such as movies) could bring new results and insights. (e)The last paragraph of the conclusion could be more discussed (possible reasons for the intriguing data presented).

[Response 1]

(a)  We appreciate the reviewer’s suggestion and we have included more details and discussions in the text regarding the analyzes. 

(b)  We basically have access to all streaming session dataset from this content provider. 

(c)  We can say that the conclusions obtained from the analyzes of those events are not particular to “big events”. However, as we consider the very popular events more critical to those enterprise companies, especially for an infrastructure forecast purpose, we focused on those events. Having said that, it is worth mentioning that we included a paragraph in the conclusion adding this and other information for the readers. 

(d)  The content provider also streams movies and series; however, at this work, we consider those out of scope, in which we have limited the analyzes to live streaming contents. But we agree that such analyzes would be great to be done for video on demand service. Thus, we added it to the text as a possibility of future work. As well, we have included new possibilities of future work of forecast using machine learning from the perspectives of our datasets. 

(e)  Again, we appreciate the suggestion. More detailed discussions regarding the observations and results have been added to this version. 

[Point 2] (a) Some information about how the authors extract the data, programming language, etc could be given. (b) It could be interesting if the dataset (or part of it) could be shared with the community.

[Response 2]

(a) Thanks for pointing this out. Indeed, such information was omitted from the original text. We have included information about technology choices and more details on this revised manuscript version. 

(b) Unfortunately, access to data is restricted for our study, as agreed with the content provider in an NDA.

[Point 3] The figures of the paper are of low quality and may be improved. The font size of the figure captions is too big. Figures 18 and 19: what is “House of representatives”, “Senate”? It is not clear. 

[Response 3]

Thanks for those observations. We have reduced the font and the figure size to increase all figures' quality.  We also improved the description of the "Live-streaming Scenario" section, in order to answer the reviewer's question.

[Point 4] There are some minor typos in the text. Ex: Line 311: mobile users have a slower recovery than de non-mobile.

[Response 4]

Thanks for pointing out and sorry for those typos. All of them have been corrected in this revised version.

Reviewer 2 Report

The authors analyze stadistics of use the mobile or non-mobile during some events of video streaming. They compare the characteristics of the fundamental metrics: Viewing Duration Time, Interruption and Reentry, Arrivals and Departures and Throughput. They present different conclusions about the analysis of data for the differents scenarios,  situation geographic, operating system of user device, etc, and speccially the behaviour of user due to interruptions of service and viewing duration.

Wrongs

Line 289: ideia
Line 158: mesure
Line 311: than de no mobile
In the description of Figure 18 you use Eurocup and Eurocopa on Figure, to use one term.

Suggestions

Why in Table 1 first apears the scenario "impeachement" in first column and in Table 2 is changed to the second. Is it would not be best to maintain order to understand it best?

Line 256. "This fact may perhaps be justified when considering that non-mobile users may have small concerns about data traffic or battery use."
Do you think that lso it could be because the costs of communication mobile and coverage and capacity of system. The wired network is more dimensioned (more capacity of users and traffic).

You indicated, in lines 291-294, some ideas over these aspects and it can be determinants.

Line 277. The localication, wired network mainly is in house or office front the connections mobile that probably are mainly in  outdoors. This can explain your conclusion in line 290: "These facts reinforce the ideia that non-mobile users are more resilient and fault-tolerant,  while mobile users are more concerned with using device resources (such as data plan and battery)".

I think that in conclusions must be indicated the high dependence of type of event for mobile connections: cost, externals aspect, movility, battery, coverage, others.

In Line 347 you separate: "Internet access as (i) mobile network, such as 3G, 4G, or equivalent, which is referred to as 3G in Table 3,and 
(ii) wired network. Since the vast majority of mobile devices have IEEE 802.11 network adapters, it was labeled Wifi". You considerate that the wired access are realized for wifi 802.11. Many conclusion can be extraided because the interface de user in a mobile device using wifi can suspend
the session and open the same session with a personal computer from the same localization (house, office). Even the case in public zones with wifi not 3G, the costs or limitations
or bandwidth (throughput per user) could be representative.

Arrivals and departures change the denomination in section 4.3  as "users in the system"

Some references could include the links and last accessed.

I think that the references could be increased. I would add  references about cases of use (different interface of user) of Internet access dependent of device and so to do some comparative or contrast.

Author Response

First of all, we thank you all for all the fruitful comments that certainly helped us to improve our paper. We have carefully considered all the remarks and suggestions made by the reviewers. We report below our answers regarding each remark raised by the reviewers and how we have addressed each remark in the revised version of the paper. It is worth mentioning that all the changes considered in the manuscript are marked in red, as suggested in the response letter, "such that changes can be easily viewed by the editors and reviewers". 

Responses to reviewers

=================================================================

Reviewer #2: The authors analyze stadistics of use the mobile or non-mobile during some events of video streaming. They compare the characteristics of the fundamental metrics: Viewing Duration Time, Interruption and Reentry, Arrivals and Departures and Throughput. They present different conclusions about the analysis of data for the differents scenarios,  situation geographic, operating system of user device, etc, and speccially the behaviour of user due to interruptions of service and viewing duration.

[Point 1] Wrongs

Line 289: ideia

Line 158: mesure

Line 311: than de no mobile

In the description of Figure 18 you use Eurocup and Eurocopa on Figure, to use one term.

[Response 1]

Thanks for pointing out and sorry for those typos. All of them have been corrected in this revised version. 

[Point 2] Why in Table 1 first apears the scenario "impeachement" in first column and in Table 2 is changed to the second. Is it would not be best to maintain order to understand it best?

[Response 2]

Thanks for the suggestion, the order of sessions in Table 1 has been changed. 

[Point 3] Line 256. "This fact may perhaps be justified when considering that non-mobile users may have small concerns about data traffic or battery use."

Do you think that lso it could be because the costs of communication mobile and coverage and capacity of system. The wired network is more dimensioned (more capacity of users and traffic).

You indicated, in lines 291-294, some ideas over these aspects and it can be determinants.

Line 277. The localication, wired network mainly is in house or office front the connections mobile that probably are mainly in  outdoors. This can explain your conclusion in line 290: "These facts reinforce the ideia that non-mobile users are more resilient and fault-tolerant,  while mobile users are more concerned with using device resources (such as data plan and battery)".

[Response 3]

We appreciate and agree with this observation. Thus, we considered this in the text trying to make it clearer.

[Point 4] I think that in conclusions must be indicated the high dependence of type of event for mobile connections: cost, externals aspect, movility, battery, coverage, others.

[Response 4]

Thanks for the suggestion. A sentence with an observation about the dependency of the results obtained in the analyzes of the results has been added to the conclusion.

[Point 5] In Line 347 you separate: "Internet access as (i) mobile network, such as 3G, 4G, or equivalent, which is referred to as 3G in Table 3, and (ii) wired network. Since the vast majority of mobile devices have IEEE 802.11 network adapters, it was labeled Wifi". You considerate that the wired access are realized for wifi 802.11. Many conclusion can be extraided because the interface de user in a mobile device using wifi can suspend the session and open the same session with a personal computer from the same localization (house, office). Even the case in public zones with wifi not 3G, the costs or limitations or bandwidth (throughput per user) could be representative.

[Response 5]

Despite the great suggestion to consider the consequences of changing the network and even the device, from the data used it is not possible to have any guess about it, since the used dataset does not have a way to identify users when they change devices and/ or network.

[Point 6] Arrivals and departures change the denomination in section 4.3 as "users in the system"

[Response 6]

Thanks for pointing that out too. S all the others, this one has also been corrected in this revised version.

[Point 7] Some references could include the links and last accessed.

I think that the references could be increased. I would add references about cases of use (different interface of user) of Internet access dependent of device and so to do some comparative or contrast.

[Response 7]

We appreciate the observation.  "Links'' and "Last accessed" information have been added to the references. And, for the references about cases of use (different interface of user) of Internet access dependent of device and so to do some comparative or contrast, we have to say that, despite our search in the short timeframe we had, we did not find other relevant works related to ours. Obviously, we will keep looking at and if we find, for sure it will be included in a possible final version.